# Predicting Childhood Obesity Using Machine Learning: Practical Considerations

**Erika R. Cheng** [1,*], **Rai Steinhardt** [2] and **Zina Ben Miled** [3,4]

1   Division of Children's Health Services Research, Department of Pediatrics, Indiana University School of Medicine, Indianapolis, IN 46202, USA
2   Purdue School of Science, Department of Computer Science, Indiana University Purdue University at Indianapolis, Indianapolis, IN 46202, USA; salemcve@iupui.edu
3   School of Engineering and Technology, Department of Electrical and Computer Engineering, Indiana University Purdue University at Indianapolis, Indianapolis, IN 46202, USA; zmiled@iupui.edu
4   Regenstrief Institute, Inc., Indianapolis, IN 46202, USA
*   Correspondence: echeng@iu.edu; Tel.: +1-317-278-0505

**Abstract:** Previous studies demonstrate the feasibility of predicting obesity using various machine learning techniques; however, these studies do not address the limitations of these methods in real-life settings where available data for children may vary. We investigated the medical history required for machine learning models to accurately predict body mass index (BMI) during early childhood. Within a longitudinal dataset of children ages 0–4 years, we developed predictive models based on long short-term memory (LSTM), a recurrent neural network architecture, using history EHR data from 2 to 8 clinical encounters to estimate child BMI. We developed separate, sex-stratified models using 80% of the data for training and 20% for external validation. We evaluated model performance using K-fold cross-validation, mean average error (MAE), and Pearson's correlation coefficient ($R^2$). Two history encounters and a 4-month prediction yielded a high prediction error and low correlation between predicted and actual BMI (MAE of 1.60 for girls and 1.49 for boys). Model performance improved with additional history encounters; improvement was not significant beyond five history encounters. The combined model outperformed the sex-stratified models, with a MAE = 0.98 (SD 0.03) and $R^2$ = 0.72. Our models show that five history encounters are sufficient to predict BMI prior to age 4 for both boys and girls. Moreover, starting from an initial dataset with more than 269 exposure variables, we were able to identify a limited set of 24 variables that can facilitate BMI prediction in early childhood. Nine of these final variables are collected once, and the remaining 15 need to be updated during each visit.

**Keywords:** childhood obesity; BMI; machine learning; EHR





## 1. Introduction

While previously uncommon in young children, obesity is now a worldwide epidemic affecting over 40 million children under the age of 5 [1,2]. Obesity in childhood is associated with both adverse outcomes like hyperlipidemia, diabetes and hypertension [3–6], as well as with higher morbidity and mortality in adulthood [7]. The underlying causes of obesity are modifiable risk factors throughout the life course; these risk factors represent major causes of health inequalities [8]. Thus, the prevention of obesity is considered a national and global health priority [9].

Unhealthy weight gain during early childhood significantly increases the risk for obesity later in life [10,11], so the ability to identify children at a young age who carry the greatest risk for obesity could significantly improve prevention efforts [12]. Several important and potentially modifiable indicators of obesity have been identified during this timeframe, including rapid infant weight gain, poor infant sleep quality, birth weight, and

maternal characteristics (e.g., current and pre-pregnancy weight, depression) [13,14]. Despite this, there has been relatively limited research into predictive modeling of childhood obesity risk, leaving many unanswered questions about how and when to intervene.

Existing research to evaluate obesity risk has predominantly employed logistic regression techniques, with limited success. The constraints of traditional regression approaches (e.g., restricting analyses to a relatively small number of predictors and assumptions of independence and linearity) have prompted others to examine non-linear interactions via machine learning [14–16]. Machine learning is increasingly recognized as useful for preventive care [17] because of its ability to characterize, adapt, learn, predict and analyze clinical data. However, one of the main challenges in employing machine learning in the clinical domain is that electronic health record (EHR) data are often incomplete and irregularly sampled (e.g., lacking regular time intervals between patient visits). In addition, height and weight, which are necessary to calculate BMI, are collected during pediatric visits in the first 2 years of life [18], but not routinely as pediatric appointments are often missed [19]. These issues hinder the performance of predictive models using EHR data. Recent techniques in deep learning and artificial neural networks address these issues and have the potential to predict health outcomes more accurately by using EHR data.

In this study, we used a longitudinal, EHR-derived dataset of children to investigate the medical history needed for a recurrent machine learning model to accurately predict BMI prior to age 4 years. Our secondary aim was to understand whether BMI prediction varies considerably between boys and girls, which would require separate BMI prediction models for each sex.

Previous studies have used machine learning techniques to develop obesity prediction models or to determine key determinants of obesity for designing intervention tools [14,20]. However, as discussed by Siddiqui et al. [20], very few of these studies analyze sex-specific prediction models, use large-scale datasets, or examine geographic/neighborhood exposure variables (e.g., access to food and opportunities for physical activity) [21,22,22–24] that might be associated with childhood obesity [25–27].

Existing models of childhood obesity risk also tend to focus on predictive variables that are routinely collected in clinical practice [28], and therefore tend to include only biological predictors and postnatal factors like infant sex and birthweight [29]. It has been suggested that one of the reasons for the intractability of childhood obesity is the failure to take into account the complexity and interconnectedness of contributing factors across the life course, ranging from the social, built, and economic environments to behavior, physiology, and epigenetics [30]. A number of childhood obesity risk factors that operate during the first 1000 days of life have been identified [13] and have special significance for obesity risk prediction. For instance, programming effects occurring during pregnancy increase children's obesity risk. Adding this information could lead to improvements in a model's ability to identify children at risk for obesity in early life, but EHR data typically contain information on maternal prenatal risk factors separately from risk factors during infancy and from measures of height and weight across childhood. The models presented in this study leverage data from a population-based, longitudinal database that combines data from multiple stages of the life course and thus add a valuable contribution to our understanding of obesity risk in early life.

Finally, the lack of effective interventions to reduce the risk for obesity in early life [31,32] suggests that efforts must be made to identify very young children with a high risk of developing obesity that could be specifically targeted for intervention. The methodology in the present paper employs long short-term memory (LSTM) [33] models to predict children's BMI prior to age 4 using different lengths of history data, determined by the number of previous clinical encounters. LSTM is a recurrent neural network model that learns from an ordered sequence of events, in this case, prior clinical encounters of the patient. While several machine learning techniques could have been used, an LSTM model was selected because the history encounter constitutes a time series. In particular, the variables height and weight that are used to calculate BMI as well as the age of the

child vary from one encounter to the next. LSTM models are particularly well suited for time-series applications and continue to outperform other architectures in various fields. For example, in Wang et al.'s analysis [34], LSTM outperformed RF, SVM, Naive Bayes, and Feed forward neural networks when predicting patient-reported outcomes using history responses from cancer patients. In other applications [35], LSTM models were used to predict post-operative risk for patients suffering from obesity and risk for complications after bariatric surgery.

## 2. Materials and Methods

### 2.1. Data Source

Data were extracted from the Obesity Prediction in Early Life (OPEL) database, a unique longitudinal, epidemiologic data repository that combines birth certificate, contextual-level, and health outcome data for 19,857 children born in Marion County, Indiana. We constructed the OPEL database by linking three independent data sources:

1.  The Child Health Improvement through Computer Automation (CHICA) system; a computer-based pediatric primary care clinical decision support system that operated in eight pediatric primary care practices in Indianapolis between 2004–2019 [36]. The CHICA system includes data for over 47,000 patients on factors such as measured height and weight, demographics (e.g., child sex, age, race/ethnicity, Medicaid insurance status), and social determinants of health (e.g., parent health literacy, food and housing insecurity, parental depression, and infant feeding practices);
2.  The IN Standard Certificate of Live Birth (i.e., 'birth certificate'), which consists of 235 variables covering parental sociodemographic information as well as information on prenatal care, labor/delivery, and neonatal conditions and procedures. Birth certificate data were made available from the Marion County Public Health Department (MCPHD); and
3.  The Social Assets and Vulnerabilities Indicators (SAVI) Project, which collects geocodes, organizes, and presents integrated data on communities in the 11-county Indianapolis metropolitan statistical area drawn from more than 30 federal, state, and local providers. All are linked to the lowest available geographic level [37]. SAVI is the nation's largest community information system, with more than 10,000 time-series variables from 1980 to the present, including welfare, education, health, public safety, housing, demographics, locations of health facilities, health and human services, community facilities, and associated service areas.

Institutional Review Board approval to construct the OPEL database was obtained from the Indiana University School of Medicine. All data analyses for this study occurred on a restricted-access server provisioned specifically for research purposes.

### 2.2. Data Preprocessing

From the OPEL database, we identified 73,957 clinical encounters from 6614 children ages 0 to 4 years. Within this limited dataset, we performed data preprocessing to remove erroneous records, impute missing values, and encode variables into normalized features for use in our predictive model. For example, encounters where height decreased more than 2 inches from the previous encounter or with implausible recorded BMIs were categorized as input error. We also established valid ranges for the mother's gestational weight gain and the child's birth weight. Variables that were one-hot encoded (e.g., race of the mother or father) were converted to multi-class nominal variables. Finally, we deleted duplicative variables, administrative variables not directly relevant to the aims of our analysis, and variables without enough data to be useful.

This preprocessing yielded a list of 269 variables derived from the OPEL database that we initially considered for modeling (Appendix A). From this list, we performed feature reduction guided by existing peer-reviewed literature on early life obesity risk (e.g., [13]), expert opinion (ERC), and the results of a LASSO regression. Feature reduction also took into account noisy and sparsely populated variables.

*2.3. Model Development*

Our outcome of interest was BMI as defined by the Center for Disease Control and Prevention (CDC) guidelines [38]. We imputed missing and invalid BMIs using linear interpolation and height and weight data from previous encounters.

After preprocessing, we randomly selected an equal number of boy and girl patients, then split the dataset by patient such that 80% of our data was used for model training and 20% was used for model testing while maintaining an equal split according to patient sex. We normalized all input variables to values between −1 and 1. In the initial dataset, the girl class was the minority class.

We then developed separate long short-term memory (LSTM) [33] models to predict BMI using different lengths of history data, determined by the number of previous clinical encounters. We defined history data as either 2, 3, 5, or 8 prior encounters, and modeled our predictions of patient BMI at each encounter immediately following the set of history encounters. We modeled predictive variables as both fixed (e.g., maternal and paternal race, infant birthweight, mother's age at birth) and varying (e.g., patient's age, visit type, sleep quality) between encounters.

The model architecture consisted of an LSTM layer followed by a single Feed forward linear layer. The number of hidden nodes in the LSTM layer was set to half the number of input features. The Adam optimizer was used to update the weights in the model. Each model was trained using an input-output sequence with a varying number of history encounters. For example, when using five history encounters the model was trained to predict BMI at the sixth encounter.

Based on prior research demonstrating different obesity determinants for boys and girls [39], we developed three models: one for boys, one for girls, and a combined model for both. K-fold cross-validation [40] with k = 5 was used to evaluate each model and to estimate variabilities induced by the data selection. The accuracy of the models was measured using MAE and Pearson's correlation coefficient ($R^2$). We report the standard deviation of these metrics from the K-fold cross-validation.

## 3. Results

The feature reduction process resulted in a set of 24 exposure variables: 15 were derived from the CHICA dataset, 7 from the birth certificate, 1 from CHICA/birth certificate, and 1 from SAVI (Table 1).

**Table 1.** Features from the OPEL database used in the analysis.

| Category | Source | Description |
|---|---|---|
| Prenatal | MCPHD | Maternal risk factor during pregnancy |
| | MCPHD | Method of delivery: vaginal versus cesarean section |
| | MCPHD | Child's birthweight (in grams) |
| Demographic | MCPHD | Child sex |
| | CHICA | Child's ethnicity |
| | CHICA | Child's age at the clinical encounter |
| | CHICA | Preferred language of the child |
| | MCPHD | Biological mother's age at delivery |
| | MCPHD | Biological mother's race and ethnicity |
| | MCPHD | Father's race and ethnicity |

**Table 1.** *Cont.*

| Category | Source | Description |
|---|---|---|
| Environmental | CHICA | Blood lead level |
| | CHICA/MCPHD | Flag for if the child has ever been enrolled in the WIC program |
| | SAVI | Percentage of the local population living in a food desert, based on child's address at birth |
| | CHICA | Parent is confident filling out health forms |
| | CHICA | Who attended the visit (e.g., mother, father, grandparent, etc.) |
| | CHICA | Flag for low health literacy risk, as determined by a validated screener |
| | CHICA | Parent response to "Are all the doors in your house that lead outside, to stairs, or potentially dangerous areas secured against [child] opening them?" |
| Developmental | CHICA | Flag for developmental delay |
| | CHICA | Parent reports concerns about the child's behavioral development |
| Sleep Quality | CHICA | Parent response to "Does [child] often wake up one or more times per night, and does an adult go to him/her?" |
| | CHICA | Parent response to "Do you think [child] has a sleep problem?" |
| Clinical | CHICA | Type of clinic visit (routine versus sick visit) |
| | CHICA | Prior BMI measurements |
| | CHICA | Time between clinical encounters |

Table 2 and Figure 1 show the distribution of the patients in the training and testing cohorts. As designed, there were approximately the same number of boys and girls included in both training and testing cohorts. There were no clinically meaningful differences across the cohorts in terms of mean BMI and age at the clinical encounter. The mean age at the encounter, defined as the average age across all encounters, was approximately 68 weeks (17 months), with no difference between the training and testing cohorts. There were also no significant differences between the cohorts with respect to the average number of encounters during the study period, although the average number of encounters for boys showed a higher standard deviation than for girls.

**Table 2.** Number of patients, average BMI, age, and number of encounters per patients included in the training and testing datasets.

| Population | N | BMI | Age at Encounter (Weeks) | Encounters per Patient * |
|---|---|---|---|---|
| | | | Mean (SD) | |
| Training Cohort | | | | |
| Male | 2694 | 16.79 (2.26) | 67.54 (57.43) | 12.56 (4.44) |
| Female | 2614 | 16.39 (2.22) | 66.75 (57.22) | 12.01 (3.69) |
| Combined | 5308 | 16.59 (2.25) | 67.16 (57.33) | 12.29 (4.10) |
| Testing Cohort | | | | |
| Male | 657 | 16.71 (2.20) | 69.07 (58.09) | 12.55 (4.18) |
| Female | 649 | 16.38 (2.20) | 67.28 (56.92) | 12.28 (4.17) |
| Combined | 1306 | 16.55 (2.21) | 68.19 (57.52) | 12.42 (4.18) |

SD, standard deviation; * Represents the average number of encounters during the timeframe of analysis.

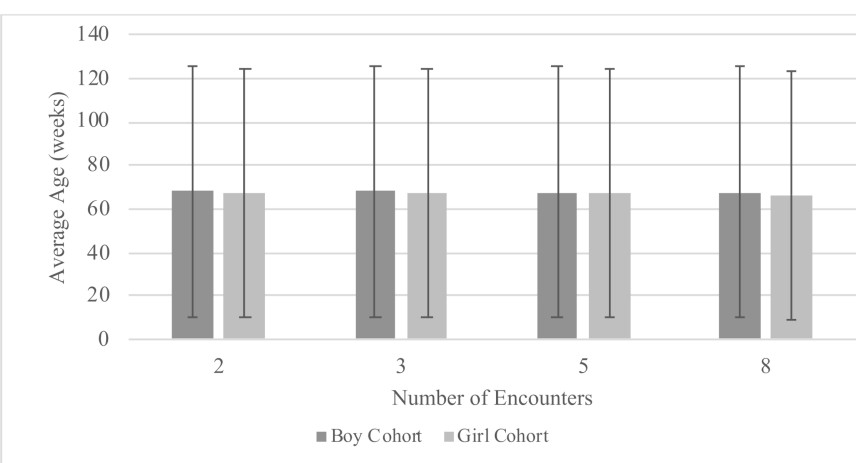

**Figure 1.** Distribution of average child age at the encounter.

Data in Table 2 were used to develop the three types of models discussed above. The boy BMI model used a total of 2694 patients during training and was tested on 657 patients. Similarly, the girl model was trained on 2614 patients and tested on 649 patients. The combined model was trained using both training cohorts (i.e., 5308 boy and girl patients) and was tested on the combined testing cohorts (i.e., 1306 boy and girl patients).

Table 3 and Figure 2 show the results of the LTSM models. Models with five or eight history encounters were determined to more accurately predict the patient's BMI than models using two or three history encounters. These models fit the observed data well, as shown by the mean average error and correlation between actual BMI and predicted BMI. Models were not trained with more than eight encounters due to concerns of reduced data quantity. Mean average error and correlation estimates were less optimal when using two or three history encounters, with the highest mean average error (1.49 for boys and 1.60 for girls) and the lowest correlation between actual and predicted BMI observed using two history encounters ($R^2 = 0.55$ in the boy only model and $R^2 = 0.49$ in the girl only model). Moreover, the K-fold standard deviation was low for both the mean average error and the $R^2$ in models with five and eight history encounters, indicating that these models were not susceptible to the selection of the training data and were more likely to generalize to new data. We observed higher K-fold standard deviations in models with two or three history encounters, suggesting less optimal performance in predicting BMI.

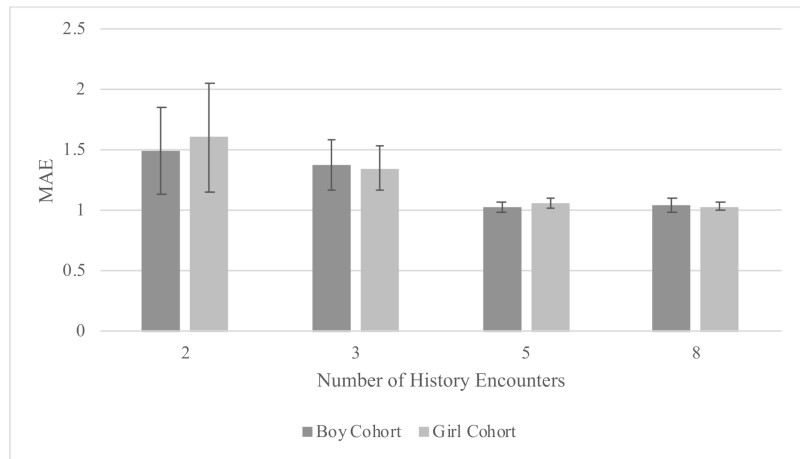

**Figure 2.** Results from the long short-term memory (LSTM) models: mean average error (MAE) by number of history encounters, stratified by child sex.

**Table 3.** Results from the long short-term memory (LSTM) models: mean average error, Pearson's correlation coefficient, and mean prediction horizon in weeks.

| History (Encounters) | MAE (SD) | $R^2$ | Prediction Horizon (Weeks) |
|---|---|---|---|
| Boy Cohort | | | |
| 8 | 1.04 (0.06) | 0.68 (0.02) | 21.56 (17.06) |
| 5 | 1.02 (0.04) | 0.68 (0.02) | 20.48 (16.87) |
| 3 | 1.37 (0.21) | 0.58 (0.07) | 18.83 (16.1) |
| 2 | 1.49 (0.36) | 0.55 (0.09) | 17.79 (15.73) |
| Girl Cohort | | | |
| 8 | 1.03 (0.03) | 0.71 (0.01) | 22.71 (17.39) |
| 5 | 1.06 (0.04) | 0.69 (0.01) | 21.18 (17.22) |
| 3 | 1.35 (0.18) | 0.62 (0.04) | 19.36 (16.37) |
| 2 | 1.60 (0.45) | 0.49 (0.14) | 18.25 (16.02) |
| Combined Cohort | | | |
| 5 | 0.98 (0.03) | 0.72 (0.01) | 20.87 (17.09) |

Each entry is the mean value of all folds in a 5 K-fold evaluation. MAE, mean average error; SD, standard deviation.

The above-mentioned advantages of the five and eight history encounter models were achieved despite having longer prediction horizons compared to the two or three history encounters models. For instance, the five history encounters boy model had an average prediction horizon of more than 20 weeks. That is, the model predicted BMI, on average, 20 weeks into the future. Conversely, the two history encounters model had an average prediction horizon of less than 18 weeks.

We did not observe significant model differences between boys and girls. The combined model showed optimal performance with the lowest mean average error (0.98, SD = 0.03) and the highest correlation ($R^2$ = 0.72), likely owing to the greater number of patients included.

Within the entire cohort, the mean age at which children reached five clinical encounters was 10.1 months with a standard deviation of 6.5 months.

## 4. Discussion

The purpose of this study was to understand the importance of historical health data in developing machine learning models to identify pediatric patients with increased risk of future overweight and obesity. Our LSTM models suggest that clinical data from at least five clinical encounters are needed to accurately predict child BMI prior to age four years with prediction horizons approximately 20 weeks in the future. In contrast to prior research [39], our combined model performed better than the models separated by sex, negating the need to develop and employ separate models for boys and girls.

Although previous studies have successfully applied machine learning to predict childhood obesity [14], few have investigated the application of these models in clinical care [28]. Our model could be employed in a pediatric clinical setting to dynamically track and predict children's BMI progression, facilitating obesity prevention through anticipatory guidance during each wellness visit. The results also suggest that having height and weight data from at least five clinical encounters may be necessary to accurately predict future BMI values. Encouragingly, the majority of patients in our sample achieved this threshold within the first 17 months of life, with 10 months being the average age at which children reached five clinical encounters. This suggests that employing our model to identify children at risk for suboptimal weight outcomes is feasible in very early childhood.

The input variables used by our model are consistent with previous findings in the literature [13]. For instance, characteristics of children's sleep such as duration, timing, and quality have been associated with obesity [41,42]. In this study, we conducted an ablation test on the two sleep quality variables (i.e., frequency of nighttime waking and parental perception of sleep quality) for the combined boys and girls model with five history encounters. The result of the ablation test shows a higher mean average error (1.03 vs. 0.98) with a larger standard deviation (0.07 vs. 0.03). The BMI correlation also dropped from 0.72 to 0.70, underscoring the important association of early sleep quality for the prediction of children's obesity risk.

Pediatricians are well-positioned to provide parents with information regarding obesity risk in early life, but many consensus guidelines recommend obesity screening in the pediatric setting only after 2 years of age when the "tipping point" of obesity onset may have already passed [43]. Further, meta-analyses indicate that BMI surveillance and counseling have only marginal effects on reducing children's BMI [44]. There is evidence that unhealthy weight gain in very early childhood of age tracks into later childhood, adolescence, and adulthood [10,11], which suggests that new approaches to help providers and parents address this problem are needed. Our screener, administered in the clinic setting, could help identify very young children at risk of unhealthy weight gain, enabling preventive counseling focused on healthy feeding, activity, and family lifestyle behaviors. Even though our findings show statistical support for postponing BMI prediction until it is possible to obtain information from five clinical encounters, the proposed models still facilitate early identification and intervention as existing guidelines recommend at least this many pediatric visits by six months of age [18]. The prediction horizon of 20 weeks and the frequency of encounters during children's first year of life means that there are numerous opportunities for providers to monitor growth, identify weight issues, and take appropriate action.

Consistent with prior research [45], the performance of our models diminished as the temporal distance between the acquisition of the exposure variables and the time of BMI prediction in the future increased. While requiring only two history encounters is attractive in practice because it enables the use of the model for a wider population, the high mean average error of the resulting predictive models makes their utility to predict obesity risk limited. The model's improvement when using five history encounters suggests that more clinical data are needed before one can correctly predict future BMI. However, further research is needed to evaluate the reproducibility and generalizability of our models before they can be applied in clinical practice for similar and related populations. Future work may wish to investigate the relative importance of the variables in our model using an external validation dataset and by conducting ablation experiments as performed in the present study for the subjective sleep quality variables.

Machine learning has been widely applied in the field of obesity research, both for the prediction of future weight outcomes and for identifying targets for intervention. Several previous studies proposed classifiers for obesity in both adults and for early childhood. For instance, Thamrin et al. [46] used linear regression and various machine learning approaches (Bayesian networks and CART models) to classify adults 18 and older as having or not having obesity based on survey data on indicators such as age, parental obesity, and activity level. Here, we predict children's future BMI rather than classify risks for obesity. We stipulate that the transparency of our proposed approach can better support intervention. Another earlier study by Dugan et al. [47] used longitudinal data from CHICA to compare different machine learning techniques (decision trees, random forest, and Bayesian networks) using 167 features from the first 2 years of life. They found that decision trees provided the best accuracy when predicting obesity between ages 2 and 10 years. Our study expands on this work by using historical data to predict children at risk for obesity. Other research focused on machine learning and obesity prediction has provided thresholds for obesity rather than BMI [48–50], which may not be as applicable for patients at younger ages. The models proposed in the present paper estimate exact BMI

values and are dynamic. They predict future BMI based on the nearest history and can therefore be used for children of varying ages. Moreover, the proposed models leverage routinely-collected EHR data, which is a practical approach compared with previous models that, for example, predict obesity using more costly and less accessible genetic data [48,51]. Importantly, the limited number of features we identify makes our model practical for use in other settings. Although the relatively narrow set of variables we identify are not all typically included in the EHR, they could be easily collected using existing screeners [28]. This data collection approach was successfully used in previous studies to obtain child birthweight and weight change between birth and 6, 9, and 12 months [52]; and to obtain data on paternal weight, maternal smoking, and breastfeeding [53].

Our study is subject to some limitations. First, it is possible that our results may be confounded by child age. While the distribution of the data (Table 2) shows that the average at encounter is approximately 68 weeks for all cohorts, patients with five or eight encounters may be older than those with two or three encounters. Their BMI may be more stable and easier to predict. This potential for confounding is the subject of a current investigation. In addition, the EHR data within the OPEL database is derived from a predominately low-income, urban population in Indianapolis, IN. Additional work in other populations is needed to externally validate our findings, as children's growth patterns may vary by socioeconomic factors [54]. Finally, we were unable to examine other variables that are potentially impactful to children's early weight gain, like physical activity, as they were not included in the OPEL database. Future research may wish to incorporate such measures for a better understanding of the children's weight trajectories.

## 5. Conclusions

The present study shows that five history encounters and a limited number of exposure variables are sufficient to predict BMI for both boys and girls in very early childhood. These findings can inform efforts to identify infants at risk of developing overweight and obesity. We envision using the proposed model in a pediatric clinic to dynamically track the progression of children's BMI four months into the future during each wellness visit. Our findings have implications for future work aimed at early identification and intervention of obesity, as well as for other chronic diseases that begin in early life.

**Author Contributions:** Conceptualization, E.R.C. and Z.B.M.; methodology, R.S. and Z.B.M.; software, Z.B.M.; validation, R.S. and Z.B.M.; formal analysis, R.S. and Z.B.M.; resources, E.R.C.; data curation, E.R.C., R.S., and Z.B.M.; writing–original draft preparation, E.R.C.; writing–review and editing, R.S. and Z.B.M.; supervision E.R.C. and Z.B.M.; funding acquisition, E.R.C. All authors have read and agreed to the published version of the manuscript.

**Funding:** This work was supported by NIH Grant K01 DK114383.

**Institutional Review Board Statement:** The study was conducted according to the guidelines of the Declaration of Helsinki and approved by the Institutional Review Board of Indiana University School of Medicine (protocol code 2006099750, approved 8 March 2020).

**Informed Consent Statement:** Not applicable because this is a retrospective study.

**Data Availability Statement:** The data presented in this study are available on request from the corresponding author. The data are not publicly available due to privacy laws.

**Acknowledgments:** The authors wish to thank Sami Gharbi for his contribution to the data acquisition and interpretation.

**Conflicts of Interest:** The authors have no conflict of interest to disclose.

**Appendix A**

Complete list of starting features before LASSO reduction by data source.

| Name | Description | Data Source |
|---|---|---|
| weight | child's weight at visit | CHICA |
| wtcentile | child's weight percentile | CHICA |
| height | child's height at visit | CHICA |
| htcentile | child's height percentile | CHICA |
| insurance | What kind of insurance, if any, the patient has at time of visit | CHICA |
| any_household_members_smoke | Do any of the people that live with the child smoke? | CHICA |
| car_seat_position_01 | Does the child use a car seat, and if so, which way is it facing? | CHICA |
| fluoride_supplemented | Does the child have fluoride supplemented somehow through consumption? | CHICA |
| has_smoke_detector | Does the child's living area have a smoke detector? | CHICA |
| hc | child's head circumference in centimeters | CHICA |
| hccentile | child's head circumference percentile | CHICA |
| know_how_to_save_choking_child | Do the child's caregivers know how to perform the Heimlich maneuver on a choking child? | CHICA |
| left_alone_in_water | Is the child left alone in water? | CHICA |
| lg_failed | What question of the language developmental test did the child fail on? | CHICA |
| maternal_depression_concern | Based on a questionnaire, is there a concern that the mom might be depressed? | CHICA |
| medicationallergies | Does the child have any medication allergies and have the allergies been confirmed by a doctor or only reported by the family? | CHICA |
| painqualitative | Is the child in pain, yes, no or NA? | CHICA |
| ps_passed | What is the highest passed question for the psychosocial developmental test? | CHICA |
| sleeps_on_side_or_back | Does the child sleep on their side or back? | CHICA |
| slept_on_stomach_ever | Does the child ever sleep on their stomach? | CHICA |
| uses_walker | Does the child use a walker? | CHICA |
| baby_left_alone_could_fall | Is the baby ever left alone where they could fall? | CHICA |
| sleeps_unsafe_soft_surface | Does the child sleep on an unsafe soft surface such as a mattress that they can suffocate on if they sleep facedown? | CHICA |
| tested_smoke_detector | If the child's living place has a smoke detector, has it been tested as working? | CHICA |
| abdomen_exam | If the child's abdomen is examined, is it abnormal or normal? | CHICA |
| back_exam | If the child's back is examined, is it abnormal or normal? | CHICA |
| chestlungs_exam | If the child's chest or lungs are examined, is it abnormal or normal? | CHICA |
| extgenitalia_exam | If the child's external genitalia is examined, is it normal or abnormal? | CHICA |
| extremities_exam | If the child's extremities (hands, feet, nose, ears) are examined, are they normal or abnormal? | CHICA |

| Name | Description | Data Source |
|---|---|---|
| fm_passed | For the fine motor skills developmental test, what is the highest question passed? | CHICA |
| general_exam | If the child had a general exam, was it normal or abnormal? | CHICA |
| gm_passed | For the gross motor skills developmental test, what is the highest passed question? | CHICA |
| head_exam | If the child's head is examined, is it normal or abnormal? | CHICA |
| heartpulses_exam | If the child's heart and pulse are examined, is it normal or abnormal? | CHICA |
| lg_passed | For the language developmental test, what is the highest scoring passed question? | CHICA |
| neuro_exam | If a neurological battery is done, was it normal or abnormal? | CHICA |
| nodes_exam | If the lymph nodes are checked, were they normal or abnormal? | CHICA |
| nosethroat_exam | If the nose and throat are examined, are they normal or abnormal? | CHICA |
| skin_exam | If the child's skin is examined, was it normal or abnormal? | CHICA |
| teethgums_exam | If the child's teeth and gums are examined, were they normal or abnormal? | CHICA |
| preferred_language | Does the child have a preferred language and if so, is it English or Spanish? | CHICA |
| burns_knowledge | Does the caregiver have knowledge of how to take care of burns? | CHICA |
| firearms_at_home | Are there any firearms in the home? | CHICA |
| firearms_where_visits | Are there any firearms where the visit is taking place? | CHICA |
| has_stairway_gates | Are there child safety gates over the stairways? | CHICA |
| household_products_out_of_reach | Are household cleaning products such as bleach out of the reach of children? | CHICA |
| matches_lighters_safe | Are matches and lighters kept in a safe manner? childproof wheel, out of reach, etc. | CHICA |
| play_area_fenced | Is the child's play area fenced in? | CHICA |
| pool_at_house | Is there a pool the child can access? | CHICA |
| chica_devscreen_status | This is a developmental screening that states whether the child is developing normally or if they are developmentally delayed and indicate which developmental screenings have been done. | CHICA |
| seen_dentist | Has the child ever been seen by a dentist? This is unlikely to be true until after the child has teeth. | CHICA |
| taking_medications | Is the child on any medications and if so, has this list of medications been confirmed to be accurate? | CHICA |
| tv_in_room | Is there a TV in the child's bedroom? | CHICA |
| tv_over_2hrs | Does the child watch TV for more than two hours every day? | CHICA |
| uses_bottle | Does the child use a bottle to eat? | CHICA |
| asthmastatus | Does the child have any asthma symptoms and if so, are they persistent, intermittent, uncontrolled/controlled? | CHICA |
| chica_devscreen_sx | Are there any developmental concerns? | CHICA |

| Name | Description | Data Source |
|---|---|---|
| lye_drain_cleaners_in_house | Are there any lye, drain, or other more dangerous cleaners in the house? | CHICA |
| ps_failed | What question of the psychosocial test did the child fail on? | CHICA |
| stop_at_curb | Does the child stop at curbs or run straight without stopping? | CHICA |
| wears_bike_helmet | Does the child wear a bike helmet for activities where one is recommended? | CHICA |
| insurancename | What kind of insurance does the child have? | CHICA |
| parents_confident_filling_out_ | Do the parents appear confident filling out forms? | CHICA |
| parents_need_help_reading | Do the parents need help reading forms? | CHICA |
| ten_childrens_books_in_home | Are there at least 10 children's books in the home available to the child? | CHICA |
| visittype | Is this a visit because the child is sick? | CHICA |
| chica_adhd_sx | Is the child having symptoms of ADHD? | CHICA |
| constipation_sx | Is the child having symptoms of constipation? | CHICA |
| firearms_kept_unloaded | Are any firearms kept unloaded in the household? | CHICA |
| look_both_ways | Does the child look both ways before crossing the street? | CHICA |
| unsupervised_near_water | Is the child left unsupervised near water? | CHICA |
| firearms_discussed | Has firearm safety been discussed with the child? | CHICA |
| grades_dropped_lately | Has the child's school grades dropped recently? | CHICA |
| knows_how_to_swim | Does the child know how to swim? | CHICA |
| rides_bike_in_street | Does the child ride their bike in the street? | CHICA |
| school_suspension_this_year | Has the child been suspended from school this year? | CHICA |
| snoring | Have parents noticed that the child snores? | CHICA |
| special_education_classes | Does the child attend special education classes? | CHICA |
| escape_plan_for_fire | Has the family discussed a house fire escape plan with their child? Older children version of smoke_alarm_knows_what_to_do | CHICA |
| informant | What household member is answering the questions? | CHICA |
| smoke_alarm_knows_what_to_do | Does the child know what to do when the smoke/fire alarm is triggered? Younger children version of escape_plan_for_fire | CHICA |
| specialneeds | Does the child have special needs or accomodations? Such as ear defenders, speech therapist, etc... | CHICA |
| visit_attendee | What household member is attending the visit but not necessarily the informant? | CHICA |
| hot_water_heater_adjusted | Has the water heater been adjusted so the water can only be heated to 120 degrees farenheit? This is a scalding concern. | CHICA |
| plastic_wrappers_secured | Are plastic wrappers in the environment secured or left in an accessible area? This is a suffocation hazard. | CHICA |
| taking_solid_food | Is the child eating solid food yet? | CHICA |
| cutting_food_bite_size | Are the child's solid foods being cut into bite size pieces before being given to the child? If no, this is a choking/suffocation hazard. | CHICA |
| carries_hot_liquids | Is the child allowed to carry hot liquids? This is a burn hazard. | CHICA |

| Name | Description | Data Source |
|---|---|---|
| play_area_cooking | Does the child have an area to play and be safely in away from cooking area while caregiver is cooking? This is a burn risk if not. | CHICA |
| safety_latches_installed | Have safety latches been installed in the house? | CHICA |
| car_seat_inspection | Has the child's car seat been inspected and if so, is it forward or rear facing? Rear facing is the safer option. | CHICA |
| developmental_referral | Has the child been referred to developmental testing and if so, have only the first steps been taken or has the appointment been made? | CHICA |
| fm_failed | What difficulty of the fine motor skills test did the child fail on? | CHICA |
| correctedvision | Does the child wear glasses or contact lenses? | CHICA |
| firearms_friends | Does the child go to friend's houses which have firearms? | CHICA |
| plays_dangerous_items | Does the child play with dangerous items? | CHICA |
| wears_sports_protective_gear | Does the child wear protective gear while playing sports? | CHICA |
| safety_caps_on_bottles | Are there child safety caps on pill bottles around the child? | CHICA |
| wears_life_jacket | Does the child wear a life jacket in situations where that is recommended? | CHICA |
| bedtime_media | Does the child use media products at bedtime? | CHICA |
| daytime_sleepiness | Is the child sleepy during the day? | CHICA |
| questionnaireinformants | Which caregiver filled out the questionnaire? | CHICA |
| sleep_quantity | Does the child get sufficient or insufficient sleep? | CHICA |
| chica_t2dm_fh | Does the child's medical records include family history? | CHICA |
| chica_t2dm_gdm | Did the child's mother have gestational diabetes? | CHICA |
| chica_t2dm_lga | Was the child large for their gestational age during pregnancy? | CHICA |
| epilepsy_history | Is there a family-reported family history of epilepsy? | CHICA |
| breast_feeding_help_needed | Does the mother need help breastfeeding? | CHICA |
| oral_exam | Has the child's mouth been examined and if so, was it normal or abnormal? | CHICA |
| bp_eval | Has the child's blood pressure been evaluated and if so, was it elevated once or repeatedly elevated? There was no option for hypotensive in this variable. | CHICA |
| empty_container_after_use | Do caregivers empty bathwater container immediately after use? This is a drowning risk if no. | CHICA |
| well_water | Does the child's household run off well-water? Well-water is a contamination concern. | CHICA |
| lowliteracyrisk | Is the child at risk of low literacy and if so, have they gone to a clinic to help? | CHICA |
| morning_headaches | Does the child have headaches in the morning or wake up with a headache? | CHICA |
| nocturnal_enuresis | Does the child wet the bed/pee during sleep? This question is for kids who are out of diapers. | CHICA |
| stops_breathing_at_night | Does the child's caregiver know if the child stops breathing during the night? | CHICA |

| Name | Description | Data Source |
|---|---|---|
| trouble_breathing_at_night | Does the child's caregiver know if the child has trouble breathing during the night? | CHICA |
| wakes_with_snort | Does the child caregiver know if the child wakes up with a snort? | CHICA |
| rides_after_dark | Does the child ride the bike after sunset? | CHICA |
| knows_rules_of_road | Does the child know traffic rules? | CHICA |
| swims_fast_moving_water | Does the child swim in fast-moving water such as a river? | CHICA |
| chica_adhd_dx | Does the child have an ADHD diagnosis? | CHICA |
| doors_secure | Are the doors in the child's home secure? | CHICA |
| sharp_edged_furniture | Are there sharp-edged furniture in the child's home? | CHICA |
| pulseox | What was the child's pulse oxygenation percentage at visit? | CHICA |
| has_window_guards | Does the child home have window guards? | CHICA |
| play_equipment_protected | Does the child play on safe playground equipment? | CHICA |
| asthmasymptoms | Does the child have symptoms of asthma? | CHICA |
| gm_failed | What gross motor test did the child fail on? | CHICA |
| chica_adhd_side_effects | Does the child experience side effects from their ADHD medication? | CHICA |
| irondeficiencyscreenqualitativ | Has the child been checked for iron deficiency and if so, what were the results? | CHICA |
| chica_devscreen_management | Is the child part of activities specifically made for children? | CHICA |
| normal_newborn_screen | Did the child have the normal newborn screen and if so, what were the results? | CHICA |
| vaccine_given | Has the child had the HPV, Tdap, or meningococcal vaccine given? | CHICA |
| anhedonia_past_few_weeks | Has the child been anhedonic/apathetic the last few weeks? | CHICA |
| cigarettes_snuff_friend | Does the child's friend or friend's household use cigarettes or snuff? | CHICA |
| cigarettes_snuff_live_with | Does someone the child lives with use snuff? | CHICA |
| ever_use_tobacco | Has the child ever used tobacco? | CHICA |
| has_drunk_alcohol | Has the child drunk alcohol at all? | CHICA |
| has_gotten_high | Has the child used an illicit substance? | CHICA |
| has_had_forced_sex_act | Has the child experienced a forced sex act? | CHICA |
| has_had_intercourse | Has the child had intercourse? | CHICA |
| sad_past_few_weeks | Has the child been sad in the past few weeks? | CHICA |
| suicide_concerns | Is there a concern of suicidality for the child? | CHICA |
| used_marijuana | Has the child used marijuana? | CHICA |
| interested_birth_control | Is the child interested in contraception? | CHICA |
| ready_to_quit | Is the child ready to quit smoking cigarettes? | CHICA |
| watches_tv | Does the child watch TV? | CHICA |
| sleep_problems | Does the child have problems sleeping? | CHICA |
| nobp | Child did not cooperate in visit; Could not check blood pressure. | CHICA |

| Name | Description | Data Source |
|---|---|---|
| nohearing | Child did not cooperate in visit; Could not perform hearing exam. | CHICA |
| risk_based_hearing_screen | Has the child undergone a hearing screen that was ordered based on high risk? | CHICA |
| chica_devscreen_treatment | Does the child have a written care plan or access to family support services? | CHICA |
| anxiety_status | Does the child have an anxiety diagnosis, or has this questionnaire been deferred? | CHICA |
| phq9_score | What was the mother's depression score on the phq9? | CHICA |
| driven_with_drunk | Has the child driven while drunk? | CHICA |
| drunk_and_activity | Has the child been drunk while doing an activity? | CHICA |
| drunk_last_month | Has the child been drunk in the last month? | CHICA |
| family_substance_abuse | Does the child's family abuse any substances? | CHICA |
| happy_how_things_going | Is the child happy with life? | CHICA |
| uses_drugs | Does the child use drugs? | CHICA |
| sudep_risk_counseling | Is the child at risk for sudden unexpected death from epilepsy? If so, is the risk high or low? | CHICA |
| surgical_hx | Has the child had their tonsils and adenoids removed? | CHICA |
| feed_at_night | Does the child eat at night? | CHICA |
| contraceptive_method_discussed | Has birth control been discussed with the child such as condoms and hormonal birth control? | CHICA |
| abuse_otc | Does the child abuse over the counter medication? | CHICA |
| abuse_steroids | Does the child abuse steroids drugs? | CHICA |
| criticized_for_drinking | Has the child been criticized for drinking? | CHICA |
| friends_use_drugs | Has the child's friends used drugs (other than alcohol/caffeine) in the last month? | CHICA |
| friend_drunk_last_month | Has the child's friends been drunk in the last month? | CHICA |
| fun_in_past_two_weeks | Does the child think they've had fun in the last two weeks? | CHICA |
| bike_has_coaster_brakes | Does the child's bike have coaster brakes? Coaster brakes allow you to pedal backwards to brake. | CHICA |
| past_depression_or_suicide | Has the child had any previous history of depression or suicidality? | CHICA |
| immune_compromise | Is the child immuno-compromised? | CHICA |
| prescription_for_cessation | Is the child on a prescribed nicotine replacement drug? | CHICA |
| intercourse_past_year | Has the child had intercourse in the last year? | CHICA |
| might_be_pregnant | Could the child be pregnant? | CHICA |
| medication | Does the child have a Ritalin prescription? | CHICA |
| depression_workup | Is there a developed safety plan for the child's depression? | CHICA |
| chica_autism_risk | Is the child at a higher risk of autism due to family history? | CHICA |
| tooth_erupted | Has the child had a tooth erupt from beneath the gums yet? | CHICA |
| autism_behavior_problems | Does the child have autism related behavior problems? | CHICA |
| autism_cam | Does the child use complementary alternative medicine for autism? | CHICA |

| Name | Description | Data Source |
|------|-------------|-------------|
| autism_financial_concerns | Are there financial concerns related to the child's autism such as paying for therapy? | CHICA |
| autism_parent_needs_respite | Is the child's caregiver in need of a break? i.e., showing symptoms of caregiver burnout | CHICA |
| patient_in_mental_health | Is the child undergoing mental health care? | CHICA |
| food_insecurity | Is the child's caregiver worried about getting enough food and if so, has this been MD confirmed or resolved? | CHICA |
| rental_status | Is the child's rental home clean & safe vs having issues, and has this been confirmed by an MD? | CHICA |
| snapdeniedlast30days | Has the child's SNAP(food stamps) been denied in the last 30 days? | CHICA |
| utility_status | Has the child's household had one of their utilities (water, power, heat, gas) shut off? Yes, no, or yes but not heat. | CHICA |
| mlp_condition_type | Is the child's family going through an eviction, on the SNAP program, or renting? | CHICA |
| wakes_up_one_or_more_times_a_n | Does the child wake up at least once during the night? | CHICA |
| wakes_up_and_needs_help_to_sleep | Does the child wake up at night and need help getting back to sleep? | CHICA |
| sleeps_on_back | Does the child sleep on their back? | CHICA |
| slept_on_stomach_side_ever | Does the child ever sleep on their stomach or side? | CHICA |
| abuse_concern | Is there a concern that the child is being abused? | CHICA |
| constipation_dx | Has the child been diagnosed with constipation? | CHICA |
| parent_thinks_child_has_sleep_pr | Do the caregivers think that the child has problems with their sleep? | CHICA |
| eyesvision_exam | Did the child have a normal or abnormal vision exam? | CHICA |
| breastfed | Is the child being breastfed at this time? | CHICA |
| psfsicklecell | Result of pre-screening form on tablet for sickle cell anemia. | CHICA |
| negativeenvironmentalhistory | Was the child potentially exposed to something negative in their environment such as tuberculosis or lead? | CHICA |
| negativenutritionhistory | Did the child have nutrition problems such as early introduction to cow milk or needing low iron formula? | CHICA |
| negativepastmedicalhistory | Did the child have a low birth weight? | CHICA |
| cholesterol_screen | Is the child at risk of high cholesterol based on parental history? | CHICA |
| earshearing_exam | Did the child have a normal or abnormal hearing exam? | CHICA |
| hearingleft | Does the child have full or partial hearing in their left ear? | CHICA |
| hearingright | Does the child have full or partial hearing in their right ear? | CHICA |
| ppd_result | What was the result of the mother's post-partum depression assessment? | CHICA |
| venousbloodleadqualitative | How much lead was in the child's blood, if tested? | CHICA |
| mother_bmi | Maternal body mass index | MCPHD |
| PNC_Clinic_Type | Type of prenatal care clinic | MCPHD |
| Sex | Child's sex | MCPHD |
| FATHER_OCCUP_DSCRP | Is child's father employed at time of birth? | MCPHD |

| Name | Description | Data Source |
|---|---|---|
| MomNativeAm | Is child's mother Native American? | MCPHD |
| Mother_Weight_Gain_P | How many pounds the mother has gained during pregnancy. | MCPHD |
| MARRIED_NOW | Are child's parents married at time of birth? | MCPHD |
| APGAR5 | Appearance, Pulse, Grimace, Activity, and Respiration at five minutes post birth. Score of 10 is good; one is bad. | MCPHD |
| BIRTH_WEIGHT_GRAM | Birth weight in grams from modern birth certificate | MCPHD |
| finalroute | How was the child delivered? | MCPHD |
| HEP_B_TEST | Was hepatitis B vaccine given at birth? | MCPHD |
| Apgar1 | Appearance, Pulse, Grimace, Activity, and Respiration at 1 min post birth. Score of 10 is good; one is bad. | MCPHD |
| Dad_Race9Eth | race of child's father | MCPHD |
| Mom_Race9Eth | race of child's mother | MCPHD |
| PREN_VISIT_NBR | number of prenatal care visits | MCPHD |
| EST_GEST | estimated gestation in weeks | MCPHD |
| MOTHER_AGE | age of the mother at birth in years | MCPHD |
| FATHER_AGE | age of the father at birth in years | MCPHD |
| PREVIOUS_LIVE_NBR | How many living babies has the mother giving birth to before? | MCPHD |
| plurality | Is this a plural or singleton birth? (twins) | MCPHD |
| BREAST_FED | Was the child breast-fed at hospital release? | MCPHD |
| MOTHER_ED | mother's education level in years | MCPHD |
| FATHER_ED | father's education level in years | MCPHD |
| LD_MECONIUM | delivery complication: was there meconium present at delivery? | MCPHD |
| LD_NONE | no delivery complications | MCPHD |
| LD_NON_VERTEX | delivery complication: child in non- vertex position | MCPHD |
| firstpnc | prenatal care initiated in first trimester | MCPHD |
| wtgrams | child's birth weight in grams | MCPHD |
| PREV_BIRTH_TOTAL | number of previous live births—all birth certificates | MCPHD |
| Kotelchuck | adequacy of prenatal care index | MCPHD |
| mdpsmoke | Did the mother smoke during pregnancy? | MCPHD |
| abcond | Were abnormal conditions present at birth? | MCPHD |
| anomaly | Was a congenital anomaly found? | MCPHD |
| infect | maternal infections | MCPHD |
| labdel | labor and delivery | MCPHD |
| mmorb | maternal morbidity | MCPHD |
| methdel | method of delivery | MCPHD |
| oblab | obstetrical labor | MCPHD |
| obproc | obstetrical procedures | MCPHD |
| risk | maternal risk factor | MCPHD |
| RACE | race of the child | CHICA |
| ETHN | ethnicity of the child | CHICA |

| Name | Description | Data Source |
|---|---|---|
| wic_ever | Has the child ever been in the WIC program? | CHICA/MCPHD |
| PERINPOVN1 | persons living in poverty as percentage of population | SAVI |
| VIOLENTN2 | violent crime (including simple assaults) per 1000 people | SAVI |
| VIOLNSTN2 | violent crime (not including simple assaults) per 1000 people | SAVI |
| AGGVASLTN2 | aggravated assaults per 1000 people | SAVI |
| ROBBERYN2 | robberies per 1000 people | SAVI |
| PROPERTYN2 | property crime per 1000 people | SAVI |
| THFTVHN2 | vehicle thefts per 1000 people | SAVI |
| BURGLARYN2 | burglaries per 1000 people | SAVI |
| WALKSCORE | walkability score | SAVI |
| FRRDTRAN1 | free and reduced lunch program participants as percentage of enrollment | SAVI |
| POVB185N1 | population below 185% poverty (proxy for reduced lunch) | SAVI |
| POVB125N1 | population below 125% poverty (proxy for free lunch) | SAVI |
| RESNEWPEN1 | total residential building permits per 100 housing units | SAVI |
| COMMALLPN1 | total commercial building permits per 100 housing units | SAVI |
| TREE_CANOPY | tree canopy as percentage of land area | SAVI |
| PCT_POP_FOOD_DESERT | percentage of population far from grocery stores | SAVI |

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
