# Peer review of "Predicting Childhood Obesity Using Machine Learning: Practical Considerations"

_biomedinformatics, doi:10.3390/biomedinformatics2010012_

Round 1

Reviewer 1 Report

Dear Authors, you did a good job constructing the longitudinal model predicting child BMI. Even though, I have a few concerns as follows and I hope you could answer them and make the manuscript more comprehensive and better delivered.

  1. The manuscript seems incomplete in places such as Data Source (line 70) and Conclusions (line 257).
  2. I hope you could showcase sample data in a table or provide a full list of variables for audience reference in supplementary materials.
  3. I hope you could include in your Methods section a general formula of how your model works.
  4. Given obesity a popular topic, your reference is on the shorter side. Could you add more references to show the recent advances in childhood obesity prediction?
  5. Are BMI related metrics, such as height in your dataset (since I see that previous weight measurements are included too)? If so, why are they in or not in the final selected features?
  6. You mentioned this model predicts children in 2-4 yrs old (line 61), while the average of you samples are in 1 yrs old (60+ wks). This could be misleading if you're referring to the 5/8 encounter cohort.
  7. Related to the above question and as you mentioned in your limitations, a sample breakdown by age/encounter groups to show the cohort heterogeneity is needed to avoid this age-encounter bias.
  8. About Prediction Horizon, what's the criteria of defining it and could you include in the Discussion section how impactful an additional 20 weeks of prognostic power is. Similarly, you have achieved an MAE < 1 and an R2 > 0.7, and how significant it is clinically? It will be perfect if you include such discussion in your manuscript.

Reviewer 2 Report

1) As you have included features from OPEL database, out of all parameters, sleep is a highly subjective parameter. Sleep disturbance in children is related to numerous factors. It would be interesting to understand the prediction model in the absence of sleep parameters. 

2) It appears the activity level of children is not considered in the model. Model of mild, moderate or high physical activity could be incorporated to understand the outcome at least in near future.

Reviewer 3 Report

Obesity is a crucial issue both for adults and children. It is associated to hyperlipidemia, diabetes, and hypertension, to which it is often the cause. An overweight during early childhood increases the risk for obesity later in life. According to literature, it is a topic of interest for several branches of the medicine, as well as all methodologies useful to predict and evaluate the associated risks.

In bioinformatics, several tools implement Machine Learning (ML) techniques to predict and to extract knowledge from large data, often unstructured. ML algorithm basing its prediction on models that may be trained or untrained. Latter may be also called supervised and unsupervised learning, respectively.

This paper introduces a general overview related to overweight and obesity issue without focusing on the state-of-the-art for ML applied to the topic of interest. Latter is missing. In my opinion, it is a crucial section within all papers.

According to manuscript, the study has been conducted on a dataset provided by Obesity Prediction in Early Life (OPEL) study which analyzed a cohort including 73,957 clinical encounters among 6,614 children that occurred between 2004 and 2019 in Indianapolis. The OPEL study mentioned at line 68 should be cited. Authors reports several information about this study but not justify the chosen, or if exists similar/other datasets.

The paper built a LSTM model to predict BMI by using “different lengths of history data”; is it a time-series?

The information that presents LSTM (lines 119-122) should be moved in introduction, and I think that this topic should be extended. For instance, the state-of-the-art could be interesting.

In Methods, authors did not provide information related to how the neural network was built (hidden layers, neurons, etc ...). Results are poor of statistical significance. In my opinion, a statistical test should be performed to check predicted values on real values, in addition to the MAE.

Authors reports (lines 185-186) “The purpose of this study was to understand the importance of historical health data in developing machine learning models to identify patients with increased risk of future overweight and obesity”. In my opinion, authors should focus on a review/survey, and not on a research article. For this purpose, I think that several methodologies should be tested and compared.

Finally, Section “Conclusions” exists but these ones are missing. I invite authors to check Section 5 (Conclusions) because it reports only the comment provided by the template. Similarly, I invite you to check: “Informed Consent Statement” (line 269), “Data Availability Statement” (line 277), “Acknowledgments” (line 282). Furthermore, author must re-generate the references because it reports two times the number of reference (e.g., “2.  2. Collaborators GO…etc…”).

Round 2

Reviewer 1 Report

Dear Authors, thank you for your revisions and they have answered most of my questions and concerns. One last point is that about the age breakdown. I would like to see at least a histogram demonstration of age distribution and how they could be stratified by encounter numbers, instead of mean + SD as in the current manuscript, since your SD is high. They can be put in supplementary files for readers' reference. Also, the "Background" in the Introduction section is a little strange in structure too. There are some double spaces in the manuscript, such as Line 186, 

Reviewer 3 Report

Dear authors, I appreciate your revisions and comments.

I suggest reporting also a dedicated plot for Table3, in order to allow a quick reading of the information. Last but not least, I suggest checking typos.

In my opinion, the manuscript is exhaustive, as well as several issues have been improved. I suggest accepting paper for publishing. Comments above are optional.
